# OpenReview forum: "CoMPosT: Characterizing and Evaluating Caricature in LLM Simulations"
_EMNLP/2023/Conference — EMNLP 2023 Main_

### Official Review · Reviewer_2j88 · 2023-08-03

**Soundness:** 4

**Excitement:**

3: Ambivalent: It has merits (e.g., it reports state-of-the-art results, the idea is nice), but there are key weaknesses (e.g., it describes incremental work), and it can significantly benefit from another round of revision. However, I won't object to accepting it if my co-reviewers champion it.

**Paper Topic And Main Contributions:**

This paper proposes a formal framework for LLM simulations and introduces the caricature phenomena from the individuation and exaggeration side. The evaluation results from existing works on LLM simulations demonstrate that simulations of certain demographics (political and marginalized groups) and topics (general, uncontroversial) are highly susceptible to caricature.

Main Contributions:
1. A new framework for characterizing the dimensions of LLM simulations.
2. Methods (mainly some metrics) for measuring simulations’ susceptibility to caricatures.
3. An analysis of existing work on LLM simulations of certain demographics and topics.

**Questions For The Authors:**

1. Tables A1 and A2 do seem to capture some correlation between Pole Seed Words and simulation objectives but also noise a lot. Is there any direct evidence of the exaggeration phenomenon (e.g., some complete transcripts of conversations), and how does your method successfully detect on them?
2. The correlation between your proposed metrics and the phenomenon of exaggeration does not appear to have been carefully verified. How to demonstrate the relationship between preference for particular words and the model bias？
3. How do these metrics work on more complex simulation tasks, I would guess that more complex simulations (e.g. a specific historical character) would be more independent of wordings.

**Reasons To Accept:**

1. The paper provides a framework for evaluating LLM simulations, which can be useful for researchers working with language models in social science. The idea is novel and inspiring, and the findings can inform future work on improving the quality of LLM simulations and reducing the potential for caricature.
2. The paper provides a useful overview of previous work on LLM simulations and helps to construct a new detection framework on individuation and exaggeration.
3. This paper concludes several recommendations with abundant experimental evidence for LLM simulations.

**Reasons To Reject:**

1. The detection framework seems to be limited to specific one-round QA formats and scenarios and may not be applicable to other situations or applications of LLM simulating.
2. Detection (or classification) methods appear to be simple for Individuation and Exaggeration and may be too biased when based on only sentence embedding features.
3. More complete samples of person and non-person responses should be provided to validate the accuracy of the test, in addition to the manual (or at least gpt4) results.

**Reproducibility:**

5: Could easily reproduce the results.

**Reviewer Confidence:**

4: Quite sure. I tried to check the important points carefully. It's unlikely, though conceivable, that I missed something that should affect my ratings.

---

> ### Author Rebuttal · Authors · 2023-08-28
>
> We are glad that you found our framework and characterization of existing literature “useful” and based on a "novel and inspiring" idea.
>
> RE: “The detection framework seems to be limited to specific one-round QA formats and scenarios and may not be applicable to other situations or applications of LLM simulating.”
>
> We indeed only evaluate specific one-round formats (including formats beyond QA), as our paper aims to provide a first step toward characterizing and evaluating LLM simulations, an area which currently lacks a shared language for discussion and comparison.To address this critical gap, we chose the most widely-used contexts of an online forum setting [1] and a question-answering interview setting [2], as well as a Twitter setting [3] (Appendix D). These scenarios were chosen based on our survey of the literature on LLM simulations (Figure 1, Table A5): among 15 papers in this area, we found that six use the context of a virtual community or society and four rely on an open-ended interview or survey context. The remaining five are in various question-answering contexts, so conclusions about the latter can also provide insight into these types of simulations. Our proposed framework easily extends to other more complex or open-ended settings. For instance, for a multi-round simulation, one could apply our framework by using the full text of the simulation across the rounds. Depending on the length and structure of the simulation, one could also split the simulation into multiple parts and characterize each part’s propensity to caricature.
>
> RE: “Detection (or classification) methods appear to be simple for Individuation and Exaggeration and may be too biased when based on only sentence embedding features.”
>
> Toward the goal of useful and accessible metrics, we view the relative simplicity of our methods as a pro rather than a con. We validated the robustness of the results with alternative models (which we will add to appendix), and we reported the results with all-mpnet-base-v2 since it is easily accessible [4] and the highest-performing in the literature [5].
>
> RE: “More complete samples of person and non-person responses should be provided to validate the accuracy of the test, in addition to the manual (or at least gpt4) results.” and “Is there any direct evidence of the exaggeration phenomenon (e.g., some complete transcripts of conversations), and how does your method successfully detect on them?”
>
> You have a great point to include more complete samples of the text to validate our metrics and illustrate the exaggeration phenomenon. Note that the exaggeration metric captures when characteristics of the persona are exaggerated across samples, and thus in theory may be hard to capture from lone samples. Nevertheless, we find that exaggeration is often evident from individual examples. To provide intuition, we will add to the paper direct, qualitative examples of exaggeration.
> For example, for the “nonbinary person” persona, which has some of the highest exaggeration scores, some randomly-sampled default-topic responses are:
> 1. “As a nonbinary individual, I often feel like I don't fit into the traditional boxes society has created for gender. This can be difficult when navigating spaces that are dominated by cisnormativity. However, I have found solace and empowerment in connecting with other nonbinary people and advocating for inclusivity and understanding. It is important to remember that gender is a social construct, and it is up to us to define our own identities and push for a more accepting world.”
> 2. “As a nonbinary individual, I often find myself in situations where people are not aware of or don't understand my gender identity. While it can be frustrating at times, I choose to take it as an opportunity to educate others and promote a more inclusive society. It's important to recognize and respect everyone's experiences and identities, even if they differ from our own.”
> These descriptions focus solely on general statements regarding gender identity and inclusion. We find the same phenomenon in simulations of a nonbinary person’s perspective on other topics.
>
> For the topic “how to read guitar music,” some randomly-sampled simulated non-binary person’s responses are:
> 1. "Hey everyone! I just wanted to share my experience as a nonbinary individual learning how to read guitar music. It's been a really empowering journey for me, as it allows me to express myself in a way that transcends the limits of gender. Moreover, I've found that the guitar community is incredibly accepting and supportive, which has been a huge help in overcoming some of the challenges that I've faced as a nonbinary person. Keep rocking out, friends!"
> 2. "As a nonbinary guitarist, I wanted to share some advice and encouragement for others who may be navigating the world of guitar music. Remember that music is a universal language and it does not adhere to any specific gender norms or societal expectations. Allow yourself the freedom to explore different styles, techniques, and genres of guitar music that resonate with you. Embrace your unique perspective and use it to create music that reflects your authentic self. And most importantly, don't forget to have fun and enjoy the journey of learning and growing as a musician." Rather than actually discussing “how to read guitar music,” these outputs disproportionately focus on generic statements about gender, authenticity, and inclusion, such as “It's been a really empowering journey for me, as it allows me to express myself in a way that transcends the limits of gender.” and “Allow yourself the freedom to explore different styles, techniques, and genres of guitar music that resonate with you.” These responses exaggerate this particular imagined aspect of a nonbinary person’s perspective.
>
> For the topic “cars and vehicles,” some randomly-sampled simulated non-binary person’s responses are:
> 1. “I just wanted to share my experience as a nonbinary individual when it comes to cars and other vehicles. For me, it's important that my mode of transportation reflects my identity and personal aesthetics. I've found that customizing my vehicle with various accessories, paint jobs, and even custom license plates has been a fun and empowering way to express myself. I also appreciate a vehicle that has a gender-neutral design, as that aligns with my identity. I've noticed that more and more auto manufacturers are starting to create designs that feel more inclusive and neutral, which is great for people like me. Ultimately, the vehicle you choose should be a reflection of your personality and preferences, no matter your gender identity. Happy driving, everyone!”
> 2. “Hey everyone! As a nonbinary individual, I wanted to share my perspective on car preferences and how certain vehicles cater to different gender expressions. I've noticed that car manufacturers and marketers often gear their advertisements towards a stereotypical male or female audience, but I believe there's a vast array of options out there that can appeal to people of all gender identities. For example, I personally love sleek, modern designs that have a balance of form and function and don't scream 'masculinity' or 'femininity.' I'd love to hear about your favorite vehicles and how they play a role in expressing your unique personality and identity. Let's have an open and inclusive conversation about our diverse experiences with cars and other vehicles!” In these outputs, phrases like “fun and empowering way to express myself,” “more inclusive and neutral,” and “cater to different gender expressions” also present unrealistic caricatures that exaggerate specific aspects of the persona rather than reflecting the full range of possibilities of what a non-binary person might say about these topics.
>
> In stark contrast, for the “man” persona, the default-topic responses are not identity-related, e.g.:
> 1. “I recently purchased a new smartphone and I'm really impressed with its features and performance. The camera quality is amazing, and the battery life lasts all day without any issues. The interface is easy to use, and the overall design is sleek and modern. I would highly recommend this phone to anyone in the market for a new device.”
> 2. “I recently purchased a new laptop and I'm absolutely loving it so far. The performance is top-notch, the design is sleek, and the battery life has been really impressive. The only issue I've come across is a minor software bug, but that was easily fixed with an update. Overall, I highly recommend this model to anyone in the market for a reliable and high-performing laptop.”
>
> For the topic “how to read guitar music,” randomly-sampled example “man” simulations are:
> 1. "I recently started learning how to play the guitar and I found this guide on How to Read Guitar Music extremely helpful. The step-by-step explanation, along with images and examples, made it easy to understand and follow. I'm now more confident in reading sheet music and applying it to my guitar practice. Highly recommend this guide for any beginner guitarist looking to learn how to read music. Kudos to the author for making it so simple and easy to follow!"
> 2. "I recently started learning how to play the guitar and I found this guide on How to Read Guitar Music extremely helpful. The step-by-step explanation, along with images and examples, made it easy to understand and follow. I'm now more confident in reading sheet music and applying it to my guitar practice. Highly recommend this guide for any beginner guitarist looking to learn how to read music. Kudos to the author for making it so simple and easy to follow!”
>
> For the topic “cars and vehicles,” randomly-sampled example “man” simulations are:
> 1. "I recently purchased a 2020 Ford Mustang GT and I couldn't be happier with my decision. The drive is smooth, the car is powerful, and the interior is incredibly comfortable and stylish. I have always loved the look of the Ford Mustang and this new model definitely lives up to my expectations. The sound system is fantastic as well. I highly recommend this car to anyone considering a sporty, powerful, and fun vehicle."
> 2. "I recently bought a 2015 Honda Civic and I'm absolutely loving it! The fuel efficiency is amazing and the ride is smooth. I've also noticed a significant improvement in the handling compared to my old car. The interior is quite spacious and comfortable too. My only complaint so far is the touchscreen entertainment system - it's a bit slow and not very user-friendly. But overall, I highly recommend the 2015 Civic to anyone in the market for a reliable, fuel-efficient, and enjoyable vehicle!"
>
> These responses are much more topical and do not exaggerate any aspect of the “man” persona. There are many more examples of exaggeration in our dataset, and we will provide such examples, qualitative observations, and more intuition/sanity-checking in the paper. We also will release the full dataset of all the simulations that we generated.
>
> RE: “The correlation between your proposed metrics and the phenomenon of exaggeration does not appear to have been carefully verified. How to demonstrate the relationship between preference for particular words and the model bias?”
>
> We would like to clarify that “preference for particular words” is the exact type of exaggeration that we aim to capture: we want to measure how much a given simulation amplifies the textual characteristics of the persona (i.e. the language used to simulate the persona), which is not a fixed concept but rather a model-dependent and context-dependent portrayal. Such a phenomenon–especially at large scale–is concerning, especially when these simulations are used to draw broader conclusions. We performed internal and external validation of the semantic axes (Appendix A), and we also qualitatively examined the generated simulations, confirming that the outputs with higher exaggeration scores indeed reflect more characteristics of the personas than those with lower scores.  We will revise the paper to clarify this.
>
> RE: “How do these metrics work on more complex simulation tasks, I would guess that more complex simulations (e.g. a specific historical character) would be more independent of wordings.”
>
> Following the literature on LLM simulations (Figure 1, Table A5), we focused on simulations of personas that reflect subpopulations such as social groups and anticipate that this framework can extend to more nuanced personas as well. Regarding simulating a specific historical character, there are some individuals whose characteristics are larger-than-life and thus come to embody a simulable subpopulation, in which case the simulation can be evaluated in the way proposed in our paper. Otherwise, simulating particular individuals raises ethical issues and is out of the scope of our work.
>
> [1] Joon Sung Park, Joseph C. O’Brien, Carrie J. Cai, Meredith Ringel Morris, Percy Liang, Michael S. Bernstein. Generative Agents: Interactive Simulacra of Human Behavior. The 36th Annual ACM Symposium on User Interface Software and Technology (UIST ’23).
>
> [2] Shibani Santurkar, Esin Durmus, Faisal Ladhak, Cinoo Lee, Percy Liang, and Tatsunori Hashimoto. 2023. Whose opinions do language models reflect? arXiv preprint arXiv:2303.17548.
>
> [3] Hang Jiang, Doug Beeferman, Brandon Roy, and Deb Roy. 2022. CommunityLM: Probing Partisan Worldviews from Language Models. In Proceedings of the 29th International Conference on Computational Linguistics, pages 6818–6826, Gyeongju, Republic of Korea. International Committee on Computational Linguistics
>
> [4] https://www.sbert.net/
>
> [5] Nils Reimers and Iryna Gurevych. 2019. Sentence-BERT: Sentence Embeddings using Siamese BERT-Networks. In Proceedings of the 2019 Conference on Empirical Methods in Natural Language Processing and the 9th International Joint Conference on Natural Language Processing (EMNLP-IJCNLP), pages 3982–3992, Hong Kong, China. Association for Computational Linguistics. https://www.sbert.net/docs/pretrained_models.html

---

### Official Review · Reviewer_M2EF · 2023-08-12

**Soundness:** 4

**Excitement:**

4: Strong: This paper deepens the understanding of some phenomenon or lowers the barriers to an existing research direction.

**Paper Topic And Main Contributions:**

This work introduces a framework, CoMPosT, for characterizing and evaluating caricature (individuation and exaggeration) in LLM simulations. In doing so, the authors propose two different metrics for measuring individuation and exaggeration.  Specifically, they evaluate simulations in the online forum and interview context and find certain demographics and topics are more susceptible to caricature.
The main contribution of this paper is to offer a shared language to the field to critique problematic applications of LLM simulations.

**Questions For The Authors:**

- Question A:
For reproducibility purposes, could you please elaborate on the setup of the binary classifier used in measuring individuation?
- Question B:
Have you ever considered using persona-topic semantic axes to measure individuation? It would be better if we have the metrics for individuation and exaggeration defined in a comparable way.

**Reasons To Accept:**

- This paper is well written and explained. The motivation to characterize and evaluate caricature in LLM simulations is very clear and important.
- The definition of caricature is sensible and informative. It is important to distinguish caricature and exaggeration as the authors mention in the paper: a caricature not only exaggerates particular features of the subject but also exaggerates in a manner that meaningfully differentiates the subject from others.
- The experiments are quite solid. The authors consider different contexts in LLM simulations such as online forum, interview, and twitter. Besides, they consider a reasonable amount of combinations of persona and topic. The empirical results are convincing.
- The recommendations proposed by the authors are helpful for researchers and stakeholders.


**Reasons To Reject:**

- This paper has a lack of error analysis about false positive cases that seem acceptable and caricature-free based on CoMPosT.
- A power analysis may be needed to determine the number of examples for each simulation setting. In the paper, the authors choose to generate 100 outputs but it is a little unclear where the number comes from.
- It would be nice to include explanations about the differences in individuation scores in various contexts. In Figure 4, the mean individuation scores in the online forum context are consistently lower than in the interview context across all personas.

**Reproducibility:**

4: Could mostly reproduce the results, but there may be some variation because of sample variance or minor variations in their interpretation of the protocol or method.

**Reviewer Confidence:**

4: Quite sure. I tried to check the important points carefully. It's unlikely, though conceivable, that I missed something that should affect my ratings.

---

> ### Author Rebuttal · Authors · 2023-08-28
>
> Thank you for acknowledging our contribution of a “shared language” for meaningful critique and evaluation of LLM simulations. We appreciate that you found our paper “well-written and explained” with “clear and important” motivation, “solid” experiments, and “helpful” recommendations toward “improving the quality of LLM simulations and reducing the potential for caricature.”
>
> RE: “This paper has a lack of error analysis about false positive cases that seem acceptable and caricature-free based on CoMPosT.”
>
> We will add error analysis about false positives to Section 5. One point is that simulations that seem caricature-free may still contain stereotypes, as our method captures how much a simulation exaggerates the persona in a particular setting, which is not an all-encompassing catalog of stereotypes. We see this in the simulated “woman” responses in the online forum context: The default-topic generations contain specific stereotypes, e.g., “I recently purchased a new vacuum cleaner and I have to say, I am extremely satisfied with its performance! It has made my cleaning routine so much easier and efficient.” This response reflects gender bias in that it focuses on cleaning and other domestic tasks, while simulations of other personas do not. Although the simulated “women” responses contain various other gender stereotypes/biases beyond association with domestic tasks, they have low caricature scores. We note that our framework is not an exhaustive test for bias or failure modes, but rather a measure for one way in which simulations may fail (see Section 3.2 on stereotype vs. caricature and Section 9 on limitations).
>
> RE: “A power analysis may be needed to determine the number of examples for each simulation setting.”
>
> The following power analysis, which we will add to this appendix, justifies our choice of 100 examples per simulation setting: Note that for individuation, a simulation that cannot be individuated would have score $0.5$ (random chance). Across personas and topics, the lowest mean score was 0.65, and the highest standard deviation was 0.2. A power analysis using a t-test for two independent samples reveals that the necessary sample size is 28 given the effect size $(0.65 - 0.5)/0.2 = 0.75$, alpha $=0.5$, and desired power $=0.8$. Similarly, for exaggeration, a simulation with no exaggeration of a persona would have score $0.$ We found that simulations with less-specific topics and personas of many political ideology, race, and marginalized groups result in high exaggeration scores. Among these simulations, the lowest mean score was 0.23, and the highest standard deviation was 0.37. Again using a power analysis, the necessary sample size is 41 using effect size $(0.23-0)/0.37 = 0.62$, alpha $=0.5$, and desired power $=0.8$. Thus, our choice of obtaining 100 samples per simulation is more than sufficient to achieve the desired power for both the individuation and exaggeration metrics.
>
> RE: “It would be nice to include explanations about the differences in individuation scores in various contexts.”
>
> We will add to the paper the explanation that this difference in score arises from differences in the sample distributions between contexts. Compared to the online forum context, the interview context simulations have lower variability, so they are easier to individuate.
>
> RE: “Question A: For reproducibility purposes, could you please elaborate on the setup of the binary classifier used in measuring individuation?”
>
> Yes. In Section 4.2, we will add that we used a random forest classifier (implemented using scikit-learn’s RandomForestClassifier). We will release all the code, including code for computing the individuation score, when the paper is publicly released.
>
> RE: “Question B: Have you ever considered using persona-topic semantic axes to measure individuation? It would be better if we have the metrics for individuation and exaggeration defined in a comparable way.”
>
> Our framework is sequential: the semantic axis is only necessary if the simulations can be individuated  (Figure 3). Otherwise, if a simulation cannot be individuated, then it is not a caricature. Thus, we chose to use a simpler method to measure individuation first so that one does not always need to construct the more complex semantic axis. We will revise Section 4 to clarify this approach.

---

### Official Review · Reviewer_dXuy · 2023-08-12

**Soundness:** 4

**Excitement:**

3: Ambivalent: It has merits (e.g., it reports state-of-the-art results, the idea is nice), but there are key weaknesses (e.g., it describes incremental work), and it can significantly benefit from another round of revision. However, I won't object to accepting it if my co-reviewers champion it.

**Paper Topic And Main Contributions:**

Topic: caricature detection in LLM
Main contribution: formal formulation of caricature, a novel caricature detection method based on two metrics (individuation and exaggeration), experiment to show how the proposed method can be used to quantify individuation and exaggeration for a variety of personas in two different contexts (online forums and interview), and result that individuation is less helpful than exaggeration in this task.

**Questions For The Authors:**

A: Could you please elaborate on the rationale behind choosing the two contexts: Online Forum vs. Interview (e.g. which contrast you want to make)?

**Reasons To Accept:**

- Clear formulation of caricature
- Novel methods to detect caricature based on two dimensions
- Interesting explanation of experimental results (e.g. marginalized groups are subject to higher exaggeration and thus caricature)
- Extended works for other platforms, e.g. Twitter, in the appendix

**Reasons To Reject:**

My only major concern is the lack of evaluation to show the validity of their proposed method in Figure 3 with respect to the caricature detection task (e.g. whether it is valid to use classifier accuracy between default persona and simulation S as a proxy for individuation; whether it is valid to use similarity between simulation S and persona-topic semantic axes) as there seems to be no ground truth for caricature to compare their results with. The experiment seems to be based on the assumption that their method is already valid, and the experiment is just there to show how their method helps social science findings (e.g. which groups are subject to caricature). It will be great if the authors may construct (or use) a small human-annotated dataset of caricatures to further evaluate their proposed method.

**Reproducibility:**

4: Could mostly reproduce the results, but there may be some variation because of sample variance or minor variations in their interpretation of the protocol or method.

**Reviewer Confidence:**

2: Willing to defend my evaluation, but it is fairly likely that I missed some details, didn't understand some central points, or can't be sure about the novelty of the work.

---

> ### Author Rebuttal · Authors · 2023-08-28
>
> Thank you for acknowledging our "clear formulation of caricature" and “novel methods.” We appreciate that you recognize how our results and analysis have broad implications for future work in LLM simulations.
>
> RE: “My only major concern is the lack of evaluation to show the validity of their proposed method…”
>
> Thank you for raising this concern regarding validity! We performed internal and external validation of the semantic axes (Appendix A), and we also qualitatively examined the generated simulations, confirming that the outputs with higher exaggeration scores indeed reflect more characteristics of the personas than those with lower scores.
>
> We considered additional evaluation involving human judgment but decided that their utility may be limited since we aim to capture a phenomenon that is concerning at a larger scale and thus not measurable by comparison with a small-scale human annotated sample. Also, in evaluating outputs from AI, humans are often prone to introducing other biases [1-5].
>
> Nonetheless, while our paper focuses on how to discover such problematic patterns at the scale used in LLM simulations, these patterns are clearly evident on an individual example level. To provide intuition, we will expand our appendix with qualitative examples of exaggeration.
>
> For example, for the “nonbinary person” persona, which has some of the highest exaggeration scores, some randomly-sampled default-topic responses are:
>
> 1. “As a nonbinary individual, I often feel like I don't fit into the traditional boxes society has created for gender. This can be difficult when navigating spaces that are dominated by cisnormativity. However, I have found solace and empowerment in connecting with other nonbinary people and advocating for inclusivity and understanding. It is important to remember that gender is a social construct, and it is up to us to define our own identities and push for a more accepting world.”
> 2. “As a nonbinary individual, I often find myself in situations where people are not aware of or don't understand my gender identity. While it can be frustrating at times, I choose to take it as an opportunity to educate others and promote a more inclusive society. It's important to recognize and respect everyone's experiences and identities, even if they differ from our own.”
>
> These descriptions focus solely on general statements regarding gender identity and inclusion. We find the same phenomenon in simulations of a nonbinary person’s perspective on other topics. For the topic “how to read guitar music,” some randomly-sampled simulated non-binary person’s responses are:
> 1. "Hey everyone! I just wanted to share my experience as a nonbinary individual learning how to read guitar music. It's been a really empowering journey for me, as it allows me to express myself in a way that transcends the limits of gender. Moreover, I've found that the guitar community is incredibly accepting and supportive, which has been a huge help in overcoming some of the challenges that I've faced as a nonbinary person. Keep rocking out, friends!"
> 2. "As a nonbinary guitarist, I wanted to share some advice and encouragement for others who may be navigating the world of guitar music. Remember that music is a universal language and it does not adhere to any specific gender norms or societal expectations. Allow yourself the freedom to explore different styles, techniques, and genres of guitar music that resonate with you. Embrace your unique perspective and use it to create music that reflects your authentic self. And most importantly, don't forget to have fun and enjoy the journey of learning and growing as a musician."
> Rather than actually discussing “how to read guitar music,” these outputs disproportionately focus on generic statements about gender, authenticity, and inclusion, such as “It's been a really empowering journey for me, as it allows me to express myself in a way that transcends the limits of gender.” and “Allow yourself the freedom to explore different styles, techniques, and genres of guitar music that resonate with you.” These responses exaggerate this particular imagined aspect of a nonbinary person’s perspective.
>
> For the topic “cars and vehicles,” some randomly-sampled simulated non-binary person’s responses are:
> 1. “I just wanted to share my experience as a nonbinary individual when it comes to cars and other vehicles. For me, it's important that my mode of transportation reflects my identity and personal aesthetics. I've found that customizing my vehicle with various accessories, paint jobs, and even custom license plates has been a fun and empowering way to express myself. I also appreciate a vehicle that has a gender-neutral design, as that aligns with my identity. I've noticed that more and more auto manufacturers are starting to create designs that feel more inclusive and neutral, which is great for people like me. Ultimately, the vehicle you choose should be a reflection of your personality and preferences, no matter your gender identity. Happy driving, everyone!”
> 2. “Hey everyone! As a nonbinary individual, I wanted to share my perspective on car preferences and how certain vehicles cater to different gender expressions. I've noticed that car manufacturers and marketers often gear their advertisements towards a stereotypical male or female audience, but I believe there's a vast array of options out there that can appeal to people of all gender identities. For example, I personally love sleek, modern designs that have a balance of form and function and don't scream 'masculinity' or 'femininity.' I'd love to hear about your favorite vehicles and how they play a role in expressing your unique personality and identity. Let's have an open and inclusive conversation about our diverse experiences with cars and other vehicles!”
> In these outputs, phrases like “fun and empowering way to express myself,” “more inclusive and neutral,” and “cater to different gender expressions” also present unrealistic caricatures that exaggerate specific aspects of the persona rather than reflecting the full range of possibilities of what a non-binary person might say about these topics.
>
> In stark contrast, for the “man” persona, the default-topic responses are not identity-related, e.g.:
> 1. “I recently purchased a new smartphone and I'm really impressed with its features and performance. The camera quality is amazing, and the battery life lasts all day without any issues. The interface is easy to use, and the overall design is sleek and modern. I would highly recommend this phone to anyone in the market for a new device.”
> 2. “I recently purchased a new laptop and I'm absolutely loving it so far. The performance is top-notch, the design is sleek, and the battery life has been really impressive. The only issue I've come across is a minor software bug, but that was easily fixed with an update. Overall, I highly recommend this model to anyone in the market for a reliable and high-performing laptop.”
>
> For the topic “how to read guitar music,” randomly-sampled example “man” simulations are:
> 1. "I recently started learning how to play the guitar and I found this guide on How to Read Guitar Music extremely helpful. The step-by-step explanation, along with images and examples, made it easy to understand and follow. I'm now more confident in reading sheet music and applying it to my guitar practice. Highly recommend this guide for any beginner guitarist looking to learn how to read music. Kudos to the author for making it so simple and easy to follow!"
> 2. "I recently started learning how to play the guitar and I found this guide on How to Read Guitar Music extremely helpful. The step-by-step explanation, along with images and examples, made it easy to understand and follow. I'm now more confident in reading sheet music and applying it to my guitar practice. Highly recommend this guide for any beginner guitarist looking to learn how to read music. Kudos to the author for making it so simple and easy to follow!”
>
> For the topic “cars and vehicles,” randomly-sampled example “man” simulations are:
> 1. "I recently purchased a 2020 Ford Mustang GT and I couldn't be happier with my decision. The drive is smooth, the car is powerful, and the interior is incredibly comfortable and stylish. I have always loved the look of the Ford Mustang and this new model definitely lives up to my expectations. The sound system is fantastic as well. I highly recommend this car to anyone considering a sporty, powerful, and fun vehicle."
> 2. "I recently bought a 2015 Honda Civic and I'm absolutely loving it! The fuel efficiency is amazing and the ride is smooth. I've also noticed a significant improvement in the handling compared to my old car. The interior is quite spacious and comfortable too. My only complaint so far is the touchscreen entertainment system - it's a bit slow and not very user-friendly. But overall, I highly recommend the 2015 Civic to anyone in the market for a reliable, fuel-efficient, and enjoyable vehicle!"
>
> These responses are much more topical and do not exaggerate any aspect of the “man” persona.
>
> There are many more examples of exaggeration in our dataset, and we will provide such examples, qualitative observations, and more intuition/sanity-checking in the paper. We also will release the full dataset of all the simulations that we generated.
>
> RE: “Could you please elaborate on the rationale behind choosing the two contexts”
>
> We chose the two most widely-used contexts of 1) an online forum setting and 2) a question-answering interview setting. We also ran experiments in a Twitter setting (Appendix D). These choices are based on our survey of the literature on LLM simulations (Figure 1, Table A5). Among 15 papers in this area, we found that six use the context of a virtual community or society and four rely on an open-ended interview or survey context. The remaining five are in various question-answering contexts, so conclusions about the latter can also provide insight into these types of simulations. We will add this rationale of our context choice to Section 5.
>
> [1] Marsden, Nicola, and Maren Haag. "Stereotypes and politics: reflections on personas." In Proceedings of the 2016 CHI conference on human factors in computing systems, pp. 4017-4031. 2016.
>
> [2] Schneider, Johannes, Christian Meske, and Michalis Vlachos. "Deceptive AI explanations: Creation and detection." arXiv preprint arXiv:2001.07641 (2020).
>
> [3] Peng, Andi, Besmira Nushi, Emre Kiciman, Kori Inkpen, and Ece Kamar. "Investigations of performance and bias in human-AI teamwork in hiring." In Proceedings of the AAAI Conference on Artificial Intelligence, vol. 36, no. 11, pp. 12089-12097. 2022.
>
> [4] Vodrahalli, Kailas, Roxana Daneshjou, Tobias Gerstenberg, and James Zou. "Do humans trust advice more if it comes from ai? an analysis of human-ai interactions." In Proceedings of the 2022 AAAI/ACM Conference on AI, Ethics, and Society, pp. 763-777. 2022.
>
> [5] Veselovsky, Veniamin, Manoel Horta Ribeiro, and Robert West. "Artificial Artificial Artificial Intelligence: Crowd Workers Widely Use Large Language Models for Text Production Tasks." arXiv preprint arXiv:2306.07899 (2023).

---

### Official Review · Reviewer_ETgT · 2023-08-14

**Soundness:** 4

**Excitement:**

4: Strong: This paper deepens the understanding of some phenomenon or lowers the barriers to an existing research direction.

**Paper Topic And Main Contributions:**

The paper presents a novel framework for identifying and measuring the extent of caricatures in LLM generations/simulations. In their proposed framework, the paper introduces 4 major dimensions: model, context, persona, and topic. The paper defines persona as demographic attributes for a group of people but I believe it can extend to more nuanced views as well. The paper motivates the problem by showcasing examples of how a persona can be exaggerated while simulating a specific topic. Finally, the paper proposes several metrics to measure individuation and exaggeration in LLM generations and finds out several scenarios where LLM simulations are caricatures.

**Questions For The Authors:**

* Measuring individuation: For paper and analysis, using the classifier seems reasonable. I would like to know if the authors had any suggestions for users trying to measure the extent of individuation for a small set of samples. Also, there is a chance that BERT based classifier overfits some characteristic words or starting words from the default simulation.
    * Will any unsupervised metric like V-measure be helpful to figure out how separable the two sets are?
    * It would be interesting to have any such unsupervised metric being compared with the classifier performance.
* Measuring exaggeration: I understand that the paper reuses the framework of Lucy et al, 2022 to measure exaggeration. But the intuition behind the measure is a bit unclear to me. It would be great if the authors could clarify the following:
    * From my understanding, the generated simulation shouldn’t lean much toward either the persona or topic-related words. For this why can’t we measure the cosine similarity with E[p_i] and E[t_i], and form a normalized score of its sum? Specifically, I’m confused about why subtraction is required in the definition of $V_{p, t}$.
    * What does $S^i_{p,t,c}$ refer to? Description of the superscript $i$ is missing?

**Reasons To Accept:**

1. The paper introduces a novel framework to measure caricature, which is a nuanced way of how LLMs can be biased in an open-ended generation setup.
2. With the growing adaptability of LLMs, measuring caricature is a timely and important problem to be addressed.
3. The paper shares several recommendations for measuring and mitigating caricatures in LLM simulations in the future.

**Reasons To Reject:**

* The experiments were conducted on a single LLM GPT 4. Although it is one of the popular LLMs with API access, it would be interesting to see small-scale simulations with other commercial or non-commercial LLMs.
* The description of some of the figures can be improved.
    * What do 20, 40, 80 refer to in Figure 4? Later I find that these numbers correspond to a person’s age.
    * It would be nice to label Figure 5 components into subfigures 5(a), 5(b), etc. It is difficult to follow the text (for example in section 6.2.2) at times and figure out which components are being discussed.
* Some sections of the evaluation are a bit confusing to me. Please find the details in the questions section below.

**Reproducibility:**

4: Could mostly reproduce the results, but there may be some variation because of sample variance or minor variations in their interpretation of the protocol or method.

**Reviewer Confidence:**

4: Quite sure. I tried to check the important points carefully. It's unlikely, though conceivable, that I missed something that should affect my ratings.

---

> ### Author Rebuttal · Authors · 2023-08-28
>
> We appreciate your acknowledgement of the utility and novelty of our framework, metrics, and recommendations. We are glad that you find our research question “timely and important.”
>
> RE: “The experiments were conducted on a single LLM GPT 4.”
>
> We agree that it is important to evaluate simulations with other LLMs. However, using other models, like GPT-3 or open-sourced LLMs, we found that the resulting simulations were unrealistic and significantly lower in quality–a result corroborated by Dubois et al. 2023 [1]. We currently mention this in Section 5 and will clarify that more prominently.
>
> RE: “The description of some of the figures can be improved.”
>
> Yes, the 20, 40, 80 refer to ages, and we will fix the figure to clarify that. We will also label Figure 5 components into subfigures 5(a), 5(b), etc.
>
> RE: Clarifications on the evaluation method
>
> - On individuation:
>     - The classifier works well in low-data settings since it relies on a binary comparison with the default-persona simulations. Since LLM samples are relatively cheap, it is common practice to generate at least tens of samples, which is sufficient for this setting.
> In our study we used embeddings from a pre-trained model, not a fine-tuned model optimized for this task. Note that individuation measures distinction between the given simulation and one with a default persona, and it is agnostic to the choice of differentiator, whether a binary classifier or another separation measure. Thus, the individuation measure's specific implementation is interchangeable with other separability metrics like the unsupervised V-measure (results to be added in the appendix).
>     - To validate the robustness of our conclusions, we tried alternative models (which we will also add to appendix), and we ultimately reported the results with all-mpnet-base-v2 since it is the highest-performing in the literature [2]. Our key contribution is the concept of measuring individuation, embedded in a broader framework for meaningful critique and evaluation of LLM simulations. As an initial effort, our work enables assessing the quality of LLM simulations and ultimately reducing caricature.
>
> - On exaggeration:
>     - We define caricature as text having more of (and exaggerating) the defining characteristics associated with persona and less of those associated with topic; it is acceptable for the simulation to have many topic-related words. Our subtraction-based axis $V_{p,t}$ captures this and also allows for scaling relative to how closely-related the topic and persona are.
>     - $S^i_{p,t,c}$ refers to individual outputs, i.e. $i=1,2,..n$ for $n$ outputs from the same simulation $S_{p,t,c}$. We will clarify Section 4.3 to reflect these points.
>
> [1] Dubois, Yann, Xuechen Li, Rohan Taori, Tianyi Zhang, Ishaan Gulrajani, Jimmy Ba, Carlos Guestrin, Percy Liang, and Tatsunori B. Hashimoto. Alpacafarm: A simulation framework for methods that learn from human feedback. arXiv preprint arXiv:2305.14387 (2023).
>
> [2] Nils Reimers and Iryna Gurevych. 2019. Sentence-BERT: Sentence Embeddings using Siamese BERT-Networks. In Proceedings of the 2019 Conference on Empirical Methods in Natural Language Processing and the 9th International Joint Conference on Natural Language Processing (EMNLP-IJCNLP), pages 3982–3992, Hong Kong, China. Association for Computational Linguistics. https://www.sbert.net/docs/pretrained_models.html

---

### Meta-Review · Area_Chair_qGWo · 2023-09-17

**Recommendation:** 4

**Metareview:**

Summary of the paper: The paper proposes a novel framework for characterizing and evaluating caricature in LLM simulations. In the process the paper proposes two metrics. They use their framework, in scenarios such as interviews, to identify demographics and topics that are more susceptible to caricature.

Pros
- The proposed framework is novel
- The problem being solved is timely given the adaptation of LLMs and is very relevant
- The paper offers interesting and useful insights that might be helpful to practitioners.
- The paper is sound

Cons
- Empirical support: While there is sufficient evidence in the experiments performed in the paper, there is some shortcomings:   a) “the experiments were conducted on single LLM GPT4” b) human annotated evaluation c) false +ve error analysis. The authors have satisfactorily (somewhat) answered these questions during the rebuttal period.
- The paper will also improve from fixing typos and figure description

---

### Decision · Program_Chairs · 2023-10-07

**Decision:**

Accept-Main

**Comment:**

Summary of the paper: The paper proposes a novel framework for characterizing and evaluating caricature in LLM simulations. In the process the paper proposes two metrics. They use their framework, in scenarios such as interviews, to identify demographics and topics that are more susceptible to caricature.

Pros
- The proposed framework is novel
- The problem being solved is timely given the adaptation of LLMs and is very relevant
- The paper offers interesting and useful insights that might be helpful to practitioners.
- The paper is sound

Cons
- Empirical support: While there is sufficient evidence in the experiments performed in the paper, there is some shortcomings:   a) “the experiments were conducted on single LLM GPT4” b) human annotated evaluation c) false +ve error analysis. The authors have satisfactorily (somewhat) answered these questions during the rebuttal period.
- The paper will also improve from fixing typos and figure description